# Road Environment Semantic Segmentation with Deep Learning from MLS Point Cloud Data

**DOI:** 10.3390/s19163466

**Published:** 2019-08-08

**Authors:** Jesús Balado, Joaquín Martínez-Sánchez, Pedro Arias, Ana Novo

**Affiliations:** Applied Geotechnologies Group, Department Natural Resources and Environmental Engineering, School of Mining and Energy Engineering, University of Vigo, Campus Lagoas-Marcosende, CP 36310 Vigo, Spain

**Keywords:** mobile mapping, PointNet, mobile laser scanning, deep learning, semantic segmentation, LiDAR, road environment

## Abstract

In the near future, the communication between autonomous cars will produce a network of sensors that will allow us to know the state of the roads in real time. Lidar technology, upon which most autonomous cars are based, allows the acquisition of 3D geometric information of the environment. The objective of this work is to use point clouds acquired by Mobile Laser Scanning (MLS) to segment the main elements of road environment (road surface, ditches, guardrails, fences, embankments, and borders) through the use of PointNet. Previously, the point cloud was automatically divided into sections in order for semantic segmentation to be scalable to different case studies, regardless of their shape or length. An overall accuracy of 92.5% has been obtained, but with large variations between classes. Elements with a greater number of points have been segmented more effectively than the other elements. In comparison with other point-by-point extraction and ANN-based classification techniques, the same success rates have been obtained for road surfaces and fences, and better results have been obtained for guardrails. Semantic segmentation with PointNet is suitable when segmenting the scene as a whole, however, if certain classes have more interest, there are other alternatives that do not need a high training cost.

## 1. Introduction

As technology evolves, cars integrate a number of sensors that make the automation of simple processes associated with driving possible, such as a brightness sensor for automatic lighting, a rain sensor for activating windscreen wipers, and a temperature sensor for a climate control system. Recently, more complex sensors have also been added to capture the environment, such as cameras and LiDAR sensors [1]. These sensors are essential for Advanced Driver Assistance Systems (ADAS) and autonomous driving. Autonomous vehicles must have knowledge at all times of its immediate environment in order to decide and interact with it.

Almost all autonomous vehicles base their perception of the built environment on LiDAR technology. LiDAR technology makes it possible to acquire the near environment of the vehicle quickly, accurately, in 3D, in true magnitude, and regardless of the light conditions. With LiDAR data, vehicles can automatically detect pedestrians [2], other vehicles [3], road signs [4], and curbs [5,6].

Some authors consider that communication between autonomous vehicles (V2V) is necessary in the near future [7]. Advantages of information exchange between vehicles are: the ability to improve the flow of traffic, minimize the risk of accidents, and optimize the process of perception of the environment, thereby minimizing specific perception errors. For example, at the end of a traffic jam, a car can indicate for others to brake without them needing to detect stopped cars; or if a car does not detect a traffic signal, this information can be provided by surrounding vehicles that have detected it. In this way, autonomous intercommunicated vehicles can be considered as a sensor network that can provide global coverage of roads. The element detection and mapping enable constant analysis of the state of roads and their immediate surroundings, schedule maintenance works actions and keep a constant record of their deterioration.

Currently, segmentation of the road environment is done by applying Artificial Intelligence (AI) techniques to images captured by vehicle cameras [8,9] and data fusion LiDAR-images [10,11,12,13]. Few authors have focused on applying AI techniques to semantic segmentation of point clouds [14,15], and heuristic methodologies are still most abundantly used [16,17,18,19]. Segmentation with point clouds is still performed by heuristic methods that focus on geometric features of point clouds that are manually identified and not directly available in images, such as geometric features, material reflectivity, and number of returns.

The objective of this work to semantically segment continuous elements in the road environment, such as road surface, guardrails, ditches, fences, embankments, and borders, through the direct use of Deep Learning applied to point clouds acquired by mobile laser scanning (MLS). The architecture of the network used is PointNet [20] and, input data are analyzed and segmented to work with constant road sections. The cloud is colored based on the reflectivity of the material and the number of returns. As a case study, the developed methodology is evaluated in a road that connects an industrial zone with a university zone.

The rest of this paper is organized as follows. Section 2 provides an overview of the classification and detection methods developed for the abovementioned elements. Section 3 presents the designed methodology. Section 4 is devoted to showing and discussing the results that were obtained. Section 5 concludes this work.

## 2. Related Work

Road conservation stakeholders need to collect data for routine and periodic maintenance works, and this task is carried out with a wide variety of procedures ranging from personnel visual inspection to aerial platforms. One challenging and current application of these inventories is developing highly-detailed (HD) and updated maps for autonomous driving. Applying MLS to road and roadside inventory has proven to be a productive approach when compared to other documentation methods [21], while also being compatible and complementary to other sensors, such as roadside surround view cameras. One of the main drawbacks of this technology is the vast amount of data acquired, which makes automation in object classification a major concern for managers and practitioners. There are a number of classification approaches in the literature based on traditional Machine Learning (ML) or Knowledge Based (KB) methods. In addition, there is increasing focus on Deep Learning (DL) techniques that may ameliorate results.

**Traditional approaches.** On the side of KB heuristics methods, the detection of obstacles based on the measurement of slope and height is studied by reference [22]. The authors work directly with 3D point clouds obtained from a stereo camera and a single-axis LiDAR. Data processing aims for autonomous navigation in cross-country environments and consists of a KB procedure. In reference [23], a Mobile Mapping System (MMS) for roads and roadside modeling are used. The approach addresses the automatic classification of a number of road assets that include lines, zebra crossings, and kerbstones based on LiDAR intensity image processing. Another KB method is applied in reference [24] to MLS data for road inventory assets such as traffic signs, trees, building walls and barriers, that are documented as a result of a first rough classification of points into three classes: ground, on-ground, and off-ground objects. LiDAR based road and pavement models are the scope of the method for curbs and curb-ramps detection proposed by [25]. In this piece of research, the procedure consists of roadside detection followed by a prediction/estimation model and a RANSAC-based filter to obtain a surface model. In reference [26], authors focus on pole-like street furniture and propose an algorithm based on a voxel regularization and look for 2D pole-section models in horizontal voxels followed by 3D voxel grouping and representation. In reference [27], a line clouds concept for road asphalt edge delineation is presented. A heuristic stage aims at structuring MLS data in lines that are afterwards grouped in accordance to a geometry-based decision tree that allows for the detection of edge points that are finally smoothed to obtain the road edges. The detection of guardrails in highways is the main objective of the work conducted by reference [28]. They combine geometric information from a Hough transform with the presence of indicators using a feature encoding method. Such features are the inputs for detection using a bag-of-features approach that relies on random-forest classification. Reference [29] focuses on the detection of corrugated guardrail based on the definition of a set of feature points obtained from models based on MLS lines. The final step of this model-based approach consists of a tracing algorithm where guardrail segments are connected and merged based on a number of conditions. In reference [30], authors investigated the fitness of Airborne Laser Scanning (ALS) for road quality assessment and performed a fiend inventory of indicators such as surface wear condition, structural condition, flatness, drying of the road, and roadside vegetation based on heuristics. A road boundary segmentation approach is presented in the work by reference [31]. Their procedure is based on MLS data and aimed to achieve real time navigation in autonomous driving. It consists of three steps based on geometry: the extraction of road boundary applying RANSAC to fitting roadside line, an Amplitude-Limiting filter, and finally a Kalman estimation filter.

Traditional ML procedures include both LiDAR and image data analysis. Images are processed in [32] to obtain 3D point clouds using Structure from Motion (SfM), that are afterwards analyzed to obtain a number of indicators of motion and 3D structure. Those 3D cues are projected to a 2D plane where road surface, roadside, fences and pole-like objects are detected, making use of a randomized forest obtained from randomized decision trees averaging. This stage results in a classifier learned from the training data where features are shared between classes. The methodology presented in reference [33] focuses on drainage ditch extraction and, to this end, authors use a random forest classifier for a first segmentation in ditch and non-ditch points followed by dropout point reconstruction using a geometric approach. In a final step, classified points are modeled as ditch objects using 2D polygons and their center lines. A classifier based on Haar-like features and Adaboost is implemented and used by reference [34] for the detection of structured barriers assets and moving elements (vehicles). Their main objective is creating a mask that would improve the results of SfM approaches for road and roadside modeling.

**Deep Learning approaches.** Image and convolutional neural networks (CNN) application to lane and road segmentation are studied in reference [35]. They deal with the comparison between precision and accuracy versus runtime and with a new mapping of the network they achieve a speed-up on KITTI dataset. Deconvolutional Networks (DN) are used by reference [36] to perform a road scene segmentation. They create what is called a Multi-Domain Road Scene Semantic Segmentation dataset from existing databases and use such a dataset to train a network without any runtime constraints and propose a transfer of its knowledge to a memory-efficient network. In reference [37], authors propose improving extensions to the aforementioned PointNet focusing on spatial context, involving neighborhood information, and exchanging information about the point descriptors over larger spatial context at output-level. The SqueezeSegV2 model [38] includes a domain transformation in order to use synthetic data from GTA-V simulators in training and generalize results to the real-world domain. DL to autonomous driving also can be applied with fisheye cameras [39], these sensors are more and more popular on vehicles. Authors propose a method called zoom augmentation for addressing the geometric models and transformation between conventional and fisheye cameras and perform the segmentation through a restricted deformable convolution (RDC), combining both real-world and transformed images. An improvement on pixel-wise semantic segmentation is detailed in reference [40]. The authors designed a dense up-sampling convolution (DUC) to achieve pixel prediction and use Cityscapes dataset for assessment. In reference [41], authors describe a CNN for urban scene classification aimed at creating HD maps for autonomous vehicles based on a Multi-Scale Voxel Network (MS3_DeepVox). End-to-end approaches for point cloud segmentation can take a multi-scale context of the points into consideration through a pyramid pooling module and recurrent neural networks (RNN) for spatial dependencies [42]. The solution was assessed on indoor and outdoor datasets. A domain adaptation approach suitable to be integrated with existing networks is presented in reference [43]. It consists of training a segmentation model to mimic a model that is pre-trained on real images. The assessment of the procedure was carried out on Cityscapes dataset.

With regard to previous approaches, this work focuses on the segmentation of all most relevant continuous elements in road environment at the same time (road surface, guardrails, ditches, fences, embankments and borders). Semantic segmentation is performed directly on the point cloud by applying Deep Learning (PointNet), without transforming it into images or using auxiliary information. The proposed approach is contrasted with a more traditional technique of point-by-point feature extraction and training with an Artificial Neural Network (ANN).

## 3. Methodology

The methodology consists of two main processes: sample generation and semantic segmentation. The input data is a point cloud of a road environment acquired with MLS and the trajectory followed by the system during the acquisition. The point cloud is segmented into point cloud sections along the trajectory to obtain regular samples with constant element distribution in road environment. Subsequently, each point cloud sample is semantically segmented by applying PointNet, identifying the points that compose each element: road surface, ditches, guardrails, fences, embankments, and borders.

### 3.1. Element Distribution in Road Environment

Urban areas are characterized by the large number of objects that are close to the road. In the case of urban corridors, the types and number of objects around the road are greatly simplified. Urban corridors have characteristics that are more typical of conventional roads and highways than of streets. The environment close to roads has very clear construction patterns (Figure 1). The road surface is the most relevant element of these environments. Road surface is a uniform and horizontal element. On each side, the road surface has a ditch that allows the evacuation of water. In addition to location, ditches are characterized by two-sided inclination in the road direction. Between ditches and road surface there may be some low vertical elements called guardrails. Guardrails protect drivers from unintentional exits from the road to prevent major accidents. Near ditches there are embankments, depending on the integration of the road with the rest of the environment. Embankments are elements of high inclination that delimit the environment of the road. Fences are vertical elements that prevent road passage from outside. Finally, the entire environment outside the road and ditch, which is not an embankment, is referred to as the border.

Although the relative location and features of these elements in the environment are clear, they are not always entirely acquired. The LiDAR position in the MLS (Figure 2) indicates that some elements are more visible than others. Road surface concentrates many points, especially in the lane where the MLS circulates. Ditches have occlusions for two reasons. One side is visible from the MLS, except when there is a guardrail between the ditch and the MLS, which partially or totally occludes it. Ditch side in contact with road surface is not visible from MLS. The same problem occurs with other elements (embankments, borders and fences) that are at a lower height than the road surface. Fences partially allow laser beam penetration, generating multiple returns in the fence net and in the elements that are behind the vehicle.

### 3.2. Sample Generation

In a complete road environment point cloud, it can be difficult to segment these elements. A complete road is an environment of variable size and shape. Segmenting road environment P(P_X_, P_Y_, P_Z_, P_I_, P_RN_, P_NR_) according to the trajectory T(T_X_, T_Y_, T_Z_) in stretches S{S_1_,S_2_, …, S_n_} of fixed size r it is possible to obtain samples where each element has a characteristic shape and location (Figure 3). For the segmentation of the road into stretches, the trajectory of the MLS during the acquisition is used [44]. Working on the trajectory saves processing time, as it contains fewer points than the entire road point cloud P, and the data is distributed linearly.

The first point (T_X1_, T_Y1_) of the path is selected as *R*_1_ and, successively, each point *R*_i_ at a distance r is stored as starting and ending points of the trajectory vectors. Then, trajectory vectors $V_{i}=\vec{R_{i}R_{i+1}}$ of each stretch are calculated. From these vectors, the perpendicular to the slope $m_{i}=-\frac{V_{Xi}}{V_{Yi}}$
in the XY plane is calculated to delimit each stretch *S_i_*. Finally, the points *P* between the perpendicular lines of each vector *V_i_* by point *R*_*i*+1_ are added to the corresponding *stretch*
$S_{i}=\{P|P_{Y}<mR_{Xi+1}-P_{X}+R_{Yi+1}\}$.

### 3.3. Semantic Segmentation

Semantic segmentation of each stretch S_i_ in point cloud is executed using PointNet [20]. PointNet is one of the first networks directly applicable to point clouds and is in the state of the art both in classification and segmentation of point clouds. PointNet has the ability to extract the geometric features between points by only using fully connected layers. The network can learn and select the most complex features (normal, eigenvalues, etc.) if they are necessary to classify points. In semantic segmentation, feature points are obtained by concatenating global and local feature vectors [45].

In addition, PoinNet allows the use of color as attributes for semantic segmentation and classification. Each stretch contains the same attributes as the original point cloud S(S_X_, S_Y_, S_Z_, S_I_, S_RN_, S_NR_). S_I_ is intensity attributes S_RN_ is return number and S_NR_ is the total number of returns (Figure 4). These three attributes are used as RGB color in PointNet input. The intensity is related to the reflectivity of the material and the elements to be detected are composed of different materials. The return number and the total number of returns are related to the penetrability of the material by laser beam. Vegetation and fences produce multiple returns. The combination of these three attributes produces information that can replace real RGB images [46].

## 4. Experiments

### 4.1. Case Study

The study area is located in the city of Vigo, Spain. It is a 3.4 km long section of the EP2005 urban corridor that connects a university zone with an industrial zone. The point cloud has been acquired with the MLS LYNX Mobile Mapper of Optech [47], is georeferenced and contains 113.6 million points. The point cloud has been hand-labeled in seven classes: road surface, ditch, guardrail, embankment, fence, border and objects. It has been automatically segmented every *r* = 5 m from the trajectory provided by the MLS during the acquisition, generating 679 samples. The size of each sample is around 10 mb and 200 million points. The number of samples has been distributed in 16% for training, 4% for validation and 80% for testing. One in five samples have been sequentially used for training/validation.

### 4.2. Training

The training has been performed on the servers provided by the Supercomputing Center of Galiza (CESGA), with an Intel^®^ Xeon^®^ E5-2680 v3 Processor, 32 GB RAM and a GPU NVidia Tesla K80. The training required 8 h. The hyper-parameters used were: optimization method Adam, Max Epochs 50, learning rate 0.001, Momentum 0.9, Batch Size 24. Figure 5 shows the accuracy and loss curve for each epoch.

### 4.3. Results

The application of the trained model in the test samples (Figure 6) obtain an overall accuracy of 92.5%, being able to classify each sample in a minute. The confusion matrix is shown in Table 1 and Table 2.

Road surface is the element that is best segmented (96.2%). Road surface is the main element of the scene, whose points are grouped and all have very similar features: inclination, planarity and number of returns, among others.

Ditch has lower accuracy (65.4%). Although the ditch location is precise, its features are not clearly identifiable: limits with road surface, edge, and slope are not always clear, nor is the existence of occlusions, changes in density, a low number of points, variations in inclination, and the depth and the existence of vegetation. These feature variations would explain its confusion with the rest of the elements. In some samples, the delimitation between road surface and ditch is precise (Figure 6a).

Although embankments also show a slight variation in slope and may contain vegetation, their segmentation has been correct (88.3%). The combination of height and inclination with respect to the road and the rest of the elements may have been relevant features.

Guardrails have had a low success rate compared to other elements. Their shape, inclination, and location should have been sufficient features for correct segmentation.

Borders are other elements that have been segmented with very good quality (95.6%). Borders have variable geometric features, may contain vegetation (even trees in some cases) and their location is not well defined, despite these difficulties its segmentation has been remarkable.

Fences have also not obtained a good segmentation (64.1%), despite the fact that they are vertical elements and their number of returns differentiates them from the other elements. The main elements with which they are generally confused are borders and embankments. In certain areas, fences are surrounded by bushes (Figure 6c). Fences can share inclination with high embankments.

In the objects class a very small number of cars and signs have been labeled. This class is not the main objective of study, and the number of samples with respect to the other elements meant that a good segmentation was not reached.

### 4.4. Comparison with Results Obtained Using ANN

Given the results obtained, and taking into account the need for very powerful equipment to train the PointNet model, the results have been evaluated in comparison to the use of an ANN by extracting features point by point.

The ANN training was performed with the same PointNet samples, from which 10,000 points were randomly selected from each class. The features extracted from each point are: intensity, return number, total number of returns, inclination [48], mean distance between points, e-values and distance to the trajectory [49]. For the features of inclination, the mean distance between points and e-values a neighborhood of 25 points has been used [50].

The neural network is composed of one hidden layer of 10 neurons. The samples have been distributed 80/10/10 for train/validation/test. Matlab has been used for extraction, training and classification. The equipment used has been an Intel i7-7700HQ CPU2.80 Hz 16 GB RAM. The training consisted of 285 epochs and has been computed in 1 min.

From each sample, features have been extracted in 2 min and the sample classified in 2 s. Table 3 shows the accuracy of the comparison between PointNet and ANN methods. Global accuracy has only reached 69.7%. The road surface and the fences have obtained the same accuracy in both methods. Ditches and objects have obtained somewhat worse results. Embankments and borders have shown a notably worse accuracy than with the use of PointNet. The only element that has improved its classification has been the guardrails.

### 4.5. Discussion

As a whole, the methodology that was implemented has achieved a good segmentation of elements of the road environment, although its low success rates in some elements not make it a viable alternative when the detection of guardrails and fences is prioritized. With respect to these two elements, there are heuristic alternatives that achieve better success rates, in particular, guardrails can best be detected with the use of an ANN and feature extraction. Road surface has achieved a very high success rate, but with a calculation of inclination and planarity of points, similar results can be obtained without the need for such expensive training. Finally, borders and embankments are complex elements that have achieved high success rates. In general, the three elements that have achieved the best segmentation are those that contain the greatest number of points and those that form the main structures of the environment. The use of PointNet in road environments is a valid option when segmenting the whole scene and not individual elements. In this case, the use of specific methodologies does not require such expensive training and better results are obtained.

## 5. Conclusions

In this work, a methodology for semantic segmentation of continuous elements (road surface, embankments, guardrails, ditches, fences and borders) conforming road environments in urban point clouds has been presented. A methodology has been implemented to segment the point cloud into sections along the road, which allows the methodology scalability regardless of road length. Semantic segmentation has been performed directly on the point cloud using PointNet. The intensity, the return number and the total number of returns have been used as false color information. The methodology has been tested in a real case study.

An overall accuracy of 92.5% has been obtained, but with a lot of variation between classes. Elements with a greater number of points (road surface, embankments, and borders) have been segmented better than the rest of elements (guardrails, ditches, and fences). The results obtained depend to a great extent on the number of points belonging to each class. In comparison with other point-by-point extraction and ANN-based classification techniques, the same success rates have been obtained for road surfaces (96%) and fences (64%), as well as very superior results in the guardrail element with an 83.6% rate (only 64.5% with PointNet). In view of the results, although PointNet allows a segmentation of the environment with good overall accuracy, in order to segment small specific elements, there are already other methodologies that achieve better success rates without the need for such costly training. Practitioners should evaluate whether to include classes that contain a relative low number of points that may be considered as noisy assets.

For future work, it would be interesting to integrate semantic segmentation based on PointNet with other heuristic methodologies that allow extracting and correcting the worst segmented elements based on best segmented elements. Another alternative would be to evaluate and implement new neural networks that improve PointNet results, such as SPLATNet [51] and KPConv [52]. Also, it is planned to increase the number of training samples with different types of road environments.

## Figures and Tables

**Figure 1 sensors-19-03466-f001:**
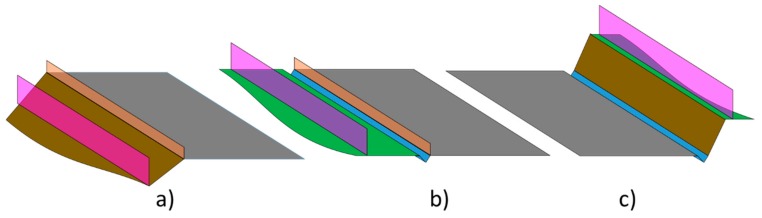
Typical configurations of elements the road environment: (**a**) road at a greater height than the environment, (**b**) road at the same height as the environment, (**c**) road at a lesser height than the environment. Color code: road surface in dark grey, ditch in blue, embankments in brown, border in green, guardrail in orange and fences in rose.

**Figure 2 sensors-19-03466-f002:**
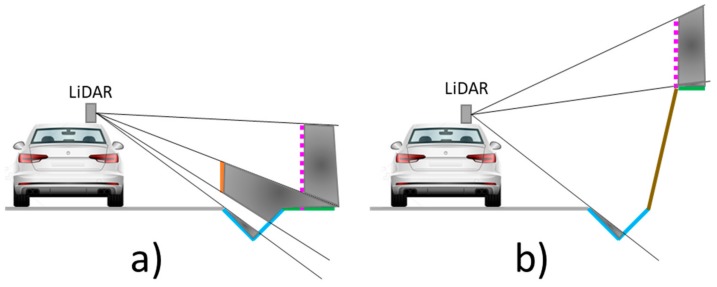
Occlusions in road elements caused by LiDAR position in roads with guardrail (**a**) and with embankments (**b**).

**Figure 3 sensors-19-03466-f003:**
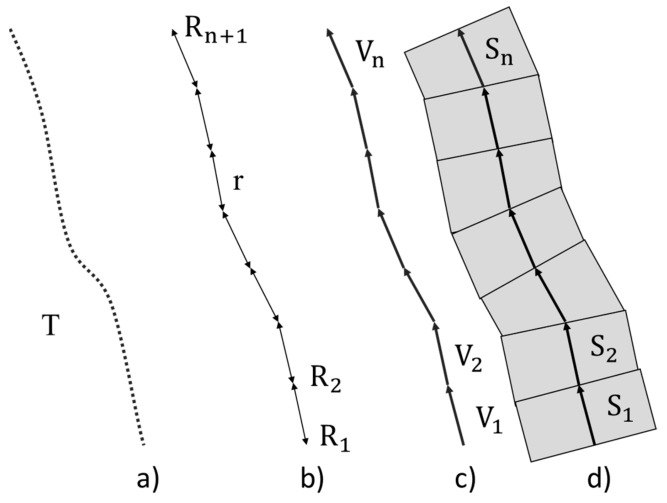
Road segmentation in stretches: (**a**) MLS trajectory points during acquisition, (**b**) selected points each distance r, (**c**) trajectory vectors, (**d**) areas of point cloud associated with each stretch.

**Figure 4 sensors-19-03466-f004:**
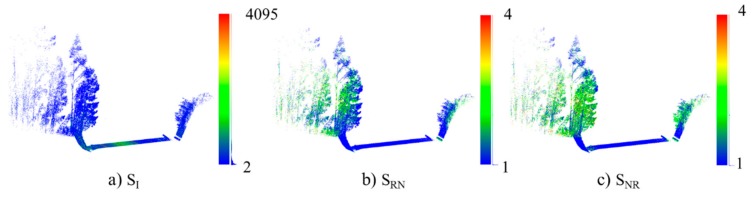
Point cloud colored by: (**a**) intensity S_I_, (**b**) return number S_RN_ and (**c**) total number of returns S_NR_.

**Figure 5 sensors-19-03466-f005:**
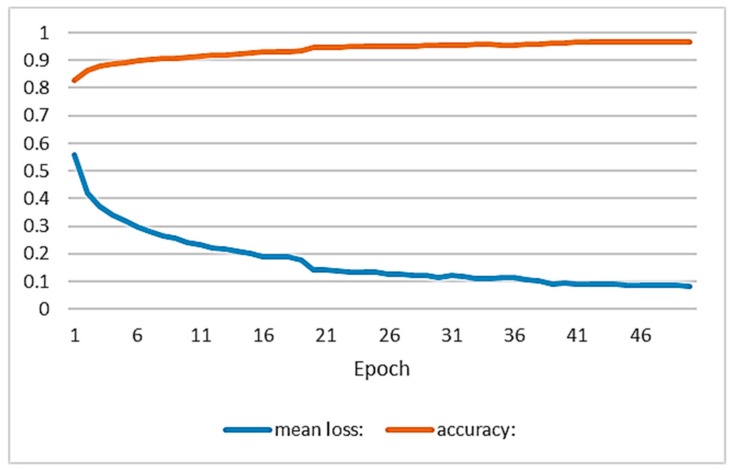
Evolution curves of accuracy (orange) and loss (blue) during training.

**Figure 6 sensors-19-03466-f006:**
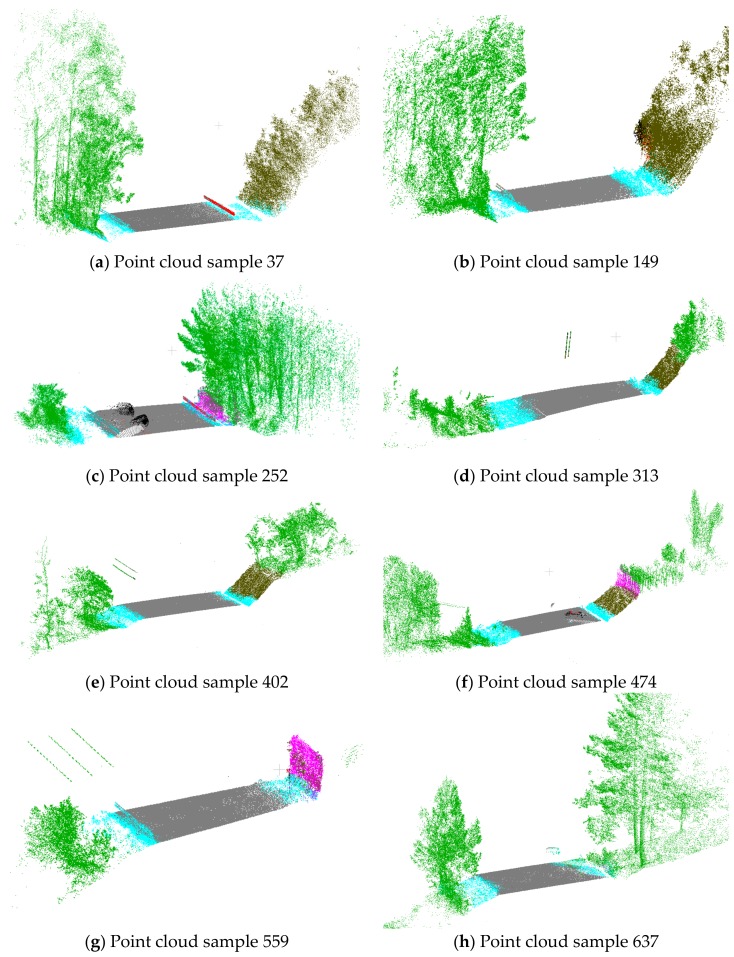
Segmented point cloud samples with PointNet. Color code: road surface in dark grey, ditch in blue, embankments in brown, border in green, guardrail in orange, and fences in rose.

**Table 1 sensors-19-03466-t001:** Confusion matrix in number of points ×10^5^.

Ref\Pred	Road S.	Ditch	Embank	Guard	Border	Fences	Objects	TOTAL
**road surface**	668.5	14.5	0.2	1.1	9.5	0.3	1.1	695.1
**ditch**	21.6	134.4	8.0	2.0	36.7	2.1	0.5	205.5
**embank.**	0.9	10.9	283.1	0.1	22.9	2.5	0.2	320.5
**guardrail**	3.7	4.8	0.1	17.9	1.0	0.1	0.1	27.7
**border**	39.2	28.3	21.1	1.1	2450.7	7.9	15.5	2563.8
**fences**	0.4	1.4	4.1	0.2	6.8	23.0	0.0	35.9
**objects**	1.7	0.9	0.2	0.6	15.5	0.8	17.5	37.3
**TOTAL**	736.0	195.2	316.8	22.9	2543.2	36.7	35.0	

**Table 2 sensors-19-03466-t002:** Confusion matrix in true positive rates.

Ref\Pred	Road Surface	Ditch	Embankments	Guardrail	Border	Fences	Objects
**road surface**	**96.2%**	2.1%	0.0%	0.2%	1.4%	0.0%	0.2%
**ditch**	10.5%	**65.4%**	3.9%	1.0%	17.9%	1.0%	0.2%
**embankments**	0.3%	3.4%	**88.3%**	0.0%	7.1%	0.8%	0.1%
**guardrail**	13.3%	17.4%	0.4%	**64.5%**	3.8%	0.3%	0.4%
**border**	1.5%	1.1%	0.8%	0.0%	**95.6%**	0.3%	0.6%
**fences**	1.1%	3.8%	11.4%	0.5%	19.0%	**64.1%**	0.1%
**objects**	4.7%	2.3%	0.6%	1.6%	41.7%	2.1%	**47.0%**

**Table 3 sensors-19-03466-t003:** Accuracy comparison between the segmentation with PointNet and ANN.

Method\Class	Road Surface	Ditch	Embankments	Guardrail	Border	Fences	Objects
**PointNet**	0.962	0.654	0.883	0.645	0.956	0.641	0.470
**ANN**	0.964	0.503	0.360	0.836	0.386	0.644	0.279

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
