# Peer review of "Road Environment Semantic Segmentation with Deep Learning from MLS Point Cloud Data"

_sensors, 2019, doi:10.3390/s19163466_

Round 1

Reviewer 1 Report

This paper shows how the authors have used PointNet as a methodology to segment road environment elements and scenes. The results have compared them using also ANN.

The topic is interesting and topical since it addresses semantic segmentation aimed at helping vehicles to know the state of the roads.

However, the paper presents certain deficiencies that must be improved:

-          The "Methodology" section should clearly state the work done in a way that is easy for the reader to understand. In the Introduction the authors explained the "why" and the motivation of their work. In this section should be presented the "what". A description of the work presented is necessary before explaining the "how".

-          The experiments are not acceptable as they are presented right now. The 92.5% of overall success that the authors claim does not agree with the partial results, since only the road and the borders have a success superior to 90% while ditches, guardrails, fences and objects have less than 65% ( in the case of objects, the success rate is 47%, which should lower the overall average considerably). It is difficult to believe that the global average is 92.5%. If so, the authors should explain how they obtain that data.

-      In addition, there is a lack of quantitative data on the number of samples in each class (we only know that there were 679 samples, but they do not comment on which samples were from each class). The confusion matrix only shows a percentage of success, instead of showing the absolute number of samples classified in each class. That information is missing and it is relevant to evaluate the system.

-          It should be justified why segmentation is semantic. The image is segmented the image to locate elements of the environment. Why are they semantic? Is any identification of elements of the environment a semantic identification?

Author Response

This paper shows how the authors have used PointNet as a methodology to segment road environment elements and scenes. The results have compared them using also ANN. The topic is interesting and topical since it addresses semantic segmentation aimed at helping vehicles to know the state of the roads. However, the paper presents certain deficiencies that must be improved:

The authors would like to thank the reviewer his contributions. All comments will be taken into account to improve the work and clarify it.

-          The "Methodology" section should clearly state the work done in a way that is easy for the reader to understand. In the Introduction the authors explained the "why" and the motivation of their work. In this section should be presented the "what". A description of the work presented is necessary before explaining the "how".

Response: Following the reviewer's comment, a paragraph has been added to the beginning of the methodology explaining what is to be done:

“The methodology consists of two main processes: sample generation and semantic segmentation. The input data is a point cloud of a road environment acquired with MLS and the trajectory during the acquisition. The point cloud is segmented into point cloud sections along the trajectory to obtain regular samples with constant element distribution in road environment. Subsequently, each point cloud sample is semantically segmented applying PointNet, identifying the points that compose each element: road surface, ditches, guardrails, fences, embankments and borders.”

-          The experiments are not acceptable as they are presented right now. The 92.5% of overall success that the authors claim does not agree with the partial results, since only the road and the borders have a success superior to 90% while ditches, guardrails, fences and objects have less than 65% (in the case of objects, the success rate is 47%, which should lower the overall average considerably). It is difficult to believe that the global average is 92.5%. If so, the authors should explain how they obtain that data.

Response: The results accounting has been carried out based on the number of points of each object correctly segmented. The elements for road surface, borders and embankments have higher success rates and a higher number of points than the others.

-      In addition, there is a lack of quantitative data on the number of samples in each class (we only know that there were 679 samples, but they do not comment on which samples were from each class). The confusion matrix only shows a percentage of success, instead of showing the absolute number of samples classified in each class. That information is missing and it is relevant to evaluate the system.

Response: The number of samples (679) refers to the number of 5m road sections in which the case study has been segmented. Each sample contains a variable percentage of points belonging to each class. Following the reviewer’s recommendation, a table has been added (new Table 1) where the number of points classified in each class is counted.

-          It should be justified why segmentation is semantic. The image is segmented the image to locate elements of the environment. Why are they semantic? Is any identification of elements of the environment a semantic identification?

Response: In a semantic segmentation, the representation of road elements includes the naming and definition of the categories and their properties and implicit relations. In our point of view, this is a semantic segmentation because the elements that compose the sample are classified point by point, therefore not only identifying the classes of elements that are present in the Point Cloud, but also localizing them and representing a relation between elements.

Reviewer 2 Report

The paper deals with road segmentation classification using deep learning (DL) techniques (PointNet) and mobile laser scanning (MLS) point cloud data. The topics is highly interesting with LIDAR and sensors of autonomous cars in backgrounds. The interesting point is also the applicability of DL with original data in comparison with traditional techniques such as ANN and feature extraction. The outcome is that DL provides better results in several hard cases, but it trades of expensive computational requirements. 

The paper is carefully written with good EN preparation. The technical details are explained clearly in details with sounded references. However, I still have several questions as follows:

1) why the distribution of train/validation/test is 16/4/80 in percents? DL is well-known to provide better results when it have more input data to train.

2) 679 generated samples is quite low number (for r=5m). Do you plan to increase the number in near future and why? What is the size of one sample, e.g., in MB?

3) why the ANN hidden layer size is chosen with 10 neurons? Does the greater number of neurons have some effect on the prediction accuracy?

4) The DL approach is able to process 1 sample per minute in deployment. What is the response of the ANN approach? I suppose very fast at the cost of manual feature extraction.

5) Do you see the real use of DL in your domain the near future and with better available hardware for sensors (e.g., GPU or FPGA), especially in real deployment to shorten the processing rate 1 sample per minute, e.g., to 1 sample per second?

Author Response

The paper deals with road segmentation classification using deep learning (DL) techniques (PointNet) and mobile laser scanning (MLS) point cloud data. The topics is highly interesting with LIDAR and sensors of autonomous cars in backgrounds. The interesting point is also the applicability of DL with original data in comparison with traditional techniques such as ANN and feature extraction. The outcome is that DL provides better results in several hard cases, but it trades of expensive computational requirements.

The paper is carefully written with good EN preparation. The technical details are explained clearly in details with sounded references. However, I still have several questions as follows:

Response: Authors would like to thank the reviewer for all the comments, which have been taken into account to improve the paper. All of them are addressed hereafter.

1) why the distribution of train/validation/test is 16/4/80 in percents? DL is well-known to provide better results when it has more input data to train.

Response: Distribution is limited by the technical resources of the equipment used for the training. The training + validation set could only be limited to 20% of the total samples. The rest has been used for testing. As the reviewer says, DL provides better results when it has more input data to train, but in our case we have not been able to use more samples for training while the distribution was fair to obtain the presented results.

2) 679 generated samples is quite low number (for r=5m). Do you plan to increase the number in near future and why? What is the size of one sample, e.g., in MB?

Response: The number of samples is low especially if we compare it with CNN image processing DL. Neural networks based on point clouds do not need such a data extension for their training nor are they as optimized as image processing DL.  However, the results achieved with a low number of samples are good.

As future work, it is planned to increase the number of samples with different types of roads, so that the methodology is applicable to more things of study. This information has been added at the end of the Conclusions: "Also, it is planned to increase the number of training samples with different types of road environments”.

The size of each sample is around 10MB and 200 million points. This information has been added at Section 4.1.

3) why the ANN hidden layer size is chosen with 10 neurons? Does the greater number of neurons have some effect on the prediction accuracy?

Response: To check if increasing the number of neurons improves the result, tests have been carried out with different numbers of neurons (10, 15, 20, 25 and 30) , where no improvements have been observed. The accuracy varies by +-2% and does not increase with the number of neurons.

4) The DL approach is able to process 1 sample per minute in deployment. What is the response of the ANN approach? I suppose very fast at the cost of manual feature extraction.

Response: Indeed, the classification with ANN is very fast, each sample is classified in approximately 2 seconds from the extracted features. But the feature extraction process takes 2 minutes. The manuscript has been completed in Section 4.4 with: “From each sample, features have been extracted in 2 minutes and the sample classified in 2 seconds.”

5) Do you see the real use of DL in your domain the near future and with better available hardware for sensors (e.g., GPU or FPGA), especially in real deployment to shorten the processing rate 1 sample per minute, e.g., to 1 sample per second?

Response: In my personal opinion, in the near future it will be possible to implement current DL techniques in vehicles, as long as there is more powerful hardware and much more refined software. We must take into account the great progress that has been made in recent years in the computer and automotive industry, with the use of new sensors and computers on board cars.

Reviewer 3 Report

The paper proposed a methodology for semantic segmentation of continuous elements (road surface, embankments, guardrails, ditches, fences and borders) conforming road environments in urban point clouds. In this paper, methodology has been implemented to segment the point cloud into sections along the road, which allows the methodology scalability regardless of road length. Semantic segmentation has been performed directly on the point cloud using PointNet. The intensity, the return number and the total number of returns have been used as false color information. The proposed methodology has been tested in a real case study, and the comparison has demonstrated the pros and cons of the proposed methodology fairly. It provided certain guiding significance for this kind of problem and had certain application value. However, there are several problems in the content of the article and the overall design of the experiment:

1.      There is a syntax error on line 15 of the article, “the point cloud is automatically divided into sections in order for semantic segmentation may be scalable to…”. It is recommended to modified.

2.      In the Section3.2, it is recommended to add the explanation of the formula and enrich the content of the PointNet in Section3.3.

3.      In the Section4.2, it is recommended to illustrate the process of training detailedly.

4.      The citation of many literatures in this paper is not standard, so it is suggested to modify

Author Response

The paper proposed a methodology for semantic segmentation of continuous elements (road surface, embankments, guardrails, ditches, fences and borders) conforming road environments in urban point clouds. In this paper, methodology has been implemented to segment the point cloud into sections along the road, which allows the methodology scalability regardless of road length. Semantic segmentation has been performed directly on the point cloud using PointNet. The intensity, the return number and the total number of returns have been used as false color information. The proposed methodology has been tested in a real case study, and the comparison has demonstrated the pros and cons of the proposed methodology fairly. It provided certain guiding significance for this kind of problem and had certain application value.

Response: The authors gratefully acknowledge to the reviewer for the careful reading and constructive comments.

However, there are several problems in the content of the article and the overall design of the experiment:

1.      There is a syntax error on line 15 of the article, “the point cloud is automatically divided into sections in order for semantic segmentation may be scalable to…”. It is recommended to modified.

Response: The sentence has been corrected: “Previously, the point cloud was automatically divided…”

2.      In the Section3.2, it is recommended to add the explanation of the formula and enrich the content of the PointNet in Section3.3.

Response: The explanation of the formula has been added in Section 4.2: “…the points P between the perpendicular lines of each vector Vi by point Ri+1 are added to the corresponding stretch Si…”.

And Section 3.3 has been extended with: “PointNet has the ability to extract the geometric features between points only by fully connected layers. The network can learn and select most complex features (normal, eigenvalues, etc.) if they are necessary to classify points. In semantic segmentation, features points are obtained by concatenating global and local feature vectors.”

3.      In the Section4.2, it is recommended to illustrate the process of training detailedly.

Response: The Section 4.2 contains information about the equipment, training time, hyperparameters and the curves of the training evolution. Sample distribution can be found at the end of Section 4.1.

4.      The citation of many literatures in this paper is not standard, so it is suggested to modify.

Response: The citation of literatures in the related work has been standardized.

Round 2

Reviewer 1 Report

The authors have improved the manuscript with several changes. However, I still believe that there are details to improve.

The new table with the number of points identified is useful because it provides data on the number of points in each class. This helps to understand why the authors get a 92.5% global success rate, since the amount of points corresponding to the road surface and the borders (where they get better succes rate) is almost an order of magnitude greater than the sum of points corresponding to fences, objects and guardrails (where they get bad classifications). It should be commented in the conclusions that this result depends a lot on the amount of points belonging to each class. It remains an open question if it is practical to include the detection of these classes or it adds more noise than useful information.

For a reason based on the comfort of the reader, I recommend that this new table also contains the sum of all the points that belong to each class. Also, since they are such high numbers, it may be more readable at 10 ^ 5 points (instead of 10 ^ 3).

Author Response

The authors have improved the manuscript with several changes. However, I still believe that there are details to improve.

Response: The authors would like to thank the reviewer for the recognition of the changes made.

The new table with the number of points identified is useful because it provides data on the number of points in each class. This helps to understand why the authors get a 92.5% global success rate, since the amount of points corresponding to the road surface and the borders (where they get better succes rate) is almost an order of magnitude greater than the sum of points corresponding to fences, objects and guardrails (where they get bad classifications). It should be commented in the conclusions that this result depends a lot on the amount of points belonging to each class. It remains an open question if it is practical to include the detection of these classes or it adds more noise than useful information.

Response: Following the recommendations of the reviewer, the second paragraph of conclusions has been completed with:

“The results obtained depend to a great extent on the number of points belonging to each class.”

And:

“Practitioners should evaluate whether to include classes that contain a relative low number of points that may be considered as noisy assets.”

For a reason based on the comfort of the reader, I recommend that this new table also contains the sum of all the points that belong to each class. Also, since they are such high numbers, it may be more readable at 10 ^ 5 points (instead of 10 ^ 3).

Response: Table 1 has been corrected according to the indications suggested by the reviewer.
